# Antidiabetic Therapy during Pregnancy: The Prescription Pattern in Italy

**DOI:** 10.3390/ijerph20237139

**Published:** 2023-12-04

**Authors:** Anna Locatelli, Sara Ornaghi, Alessandra Terzaghi, Valeria Belleudi, Filomena Fortinguerra, Francesca Romana Poggi, Serena Perna, Francesco Trotta

**Affiliations:** 1School of Medicine and Surgery, University of Milano-Bicocca, Via Pergolesi 33, 20900 Monza, Italy; anna.locatelli@unimib.it (A.L.); a.terzaghi1@campus.unimib.it (A.T.); 2Department of Epidemiology, Lazio Regional Health Service, ASL Roma 1, 00147 Rome, Italy; v.belleudi@deplazio.it (V.B.); f.poggi@deplazio.it (F.R.P.); 3Italian Medicines Agency (AIFA), 00187 Rome, Italy; f.fortinguerra@aifa.gov.it (F.F.); s.perna@aifa.gov.it (S.P.); f.trotta@aifa.gov.it (F.T.)

**Keywords:** gestational diabetes mellitus, pregestational diabetes mellitus, pregnancy, antidiabetic drugs, prescription pattern

## Abstract

Pregestational and gestational diabetes mellitus are relevant complications of pregnancy, and antidiabetic drugs are prescribed to obtain glycemic control and improve perinatal outcomes. The objective of this study was to describe the prescription pattern of antidiabetics before, during and after pregnancy in Italy and to evaluate its concordance with the Italian guideline on treatment of diabetes mellitus. A multi-database cross-sectional population study using a Common Data Model was performed. In a cohort of about 450,000 women, the prescribing profile of antidiabetics seemed to be in line with the Italian guideline, which currently does not recommend the use of oral antidiabetics and non-insulin injection, even if practice is still heterogeneous (up to 3.8% in the third trimester used oral antidiabetics). A substantial variability in the prescription pattern was observed among the Italian regions considered: the highest increase was registered in Tuscany (4.2%) while the lowest was in Lombardy (1.5%). Women with multiple births had a higher proportion of antidiabetic prescriptions than women with singleton births both in the preconception period and during pregnancy (1.3% vs. 0.7%; 3.4% vs. 2.6%) and used metformin more frequently. The consumption of antidiabetics in foreign women was higher than Italians (second trimester: 1.8% vs. 0.9%, third trimester: 3.6% vs. 1.8%).

## 1. Introduction

Hyperglycemia is one of the most prevalent medical conditions during pregnancy. According to the estimates of the International Diabetes Federation (IDF), approximately one in six live births (16.8%) is attributed to women experiencing some form of hyperglycemia during pregnancy [1]. Elevated blood glucose levels detected for the first time during pregnancy must be classified either as gestational diabetes mellitus (GDM) or as diabetes mellitus (DM) in pregnancy. In 16% of cases, hyperglycemia results from pre-existing diabetes that was already known before pregnancy or has been first identified during the pregnancy itself, while in the remaining 84%, hyperglycemia is attributed to GDM. Both these forms of diabetes are rising due to increasing risk factors in the obstetric population, such as obesity, advanced maternal age and metabolic syndrome.

Pregestational DM is a condition of variable severity of carbohydrate intolerance present before pregnancy. This definition encompasses both type 1 DM (T1DM) and type 2 DM (T2DM), already diagnosed and known before pregnancy, as well as diabetes manifested for the first time during pregnancy. This last form of diabetes has equivalent diagnostic criteria to those used for the diagnosis of T2DM outside of pregnancy: a fasting blood glucose ≥ 126 mg/dL, a random blood glucose ≥ 200 mg/dL (subsequently confirmed by fasting blood glucose ≥ 126 mg/dL) or a glycosylated hemoglobin (HbA1c) ≥ 6.5%. Regardless of the method used, results exceeding the normal range should be confirmed in a second blood sample.

Gestational diabetes mellitus (GDM) is defined as any degree of glucose intolerance with onset during pregnancy. Pregnancy-related changes in hormone balance and weight gain produce a decreased response to insulin. In most pregnancies, insulin production is adequate to overcome this resistance; however, in some circumstances, this does not occur, thus leading to the onset of GDM [2]. In Italy, screening of GDM involves the use of a 75 g glucose load curve. This test can be prescribed by physicians on maternal risk factors [3] or, as the AMD-SID group proposed, in all women between the 24th and 28th gestational weeks [4]. Women with one or more plasma glucose values exceeding the threshold are defined as having gestational diabetes: fasting ≥ 92 mg/dL, after 1 h ≥ 180 mg/dL and/or after 2 h ≥ 153 mg/dL. Usually, this condition resolves after childbirth, but it can recur years later as T2DM.

Due to the progressive increase in the age of women at childbirth and in the rate of obesity and chronic diseases, GDM has become one of the most common complications diagnosed during gestation. According to Italian and European prevalence data, approximately 6–7% of all pregnancies are complicated by GDM every year, which counts for more than 40,000 new GDM diagnoses yearly in Italy [5]. Notably, GDM is associated with increased maternal and perinatal morbidity and mortality, as well as long-term complications in both the mother and her offspring, thus representing a severe global public health issue in the world as well as in Italy [6].

The targeted monitoring and treatment of diabetes in pregnancy is required to minimize the occurrence of adverse outcomes [7,8,9]. Appropriate management of diabetes during pregnancy reduces the risk of preeclampsia, excessive maternal gestational weight gain, abortions, fetal malformation, fetal macrosomia, severe perineal injury, shoulder dystocia and neonatal hypoglycemia. Also, adequate management has a positive impact on potential maternal long-term metabolic adverse consequences, such as impaired glucose tolerance, T2DM or elevated Body Mass Index (BMI).

A targeted diet and physical activity represent the first-line interventions to obtain optimal glycemic control in pregnant women diagnosed with diabetes. However, the definition of what represents an unsuccessful attempt at a targeted diet and exercise has not been established [10,11]. Consequently, the need to start pharmacotherapy is at the specialist’s discretion, with wide variability in practice [12,13]. Both insulin and its analogues and oral antidiabetic drugs, including metformin and glibencamide, can be safely prescribed during pregnancy [14,15]. Two recent systematic reviews have found similar effectiveness of both compounds [16,17]. Yet, given the discrepancies in glycemic goals in different settings, patients’ compliance with treatment and lack of data on long-term outcomes of oral antidiabetics in offspring, a broadly accepted consensus regarding the optimal approach to pharmacological treatment of diabetes during pregnancy is still lacking.

The American College of Obstetricians and Gynecologists (ACOG) [18] and the American Diabetes Association (ADA) [19] support the use of insulin as first line therapy; notwithstanding this, they also approve the use of oral antidiabetics. The ACOG specifically suggests the use of an oral agent when a woman declines insulin therapy or is deemed incapable of safely self-administering insulin. Among oral antidiabetic agents, metformin is to be preferred over glibencamide. According to ADA guidelines, metformin should not be prescribed to women with hypertension, preeclampsia or those at risk of intrauterine growth restriction due to its potential effects on fetal growth and the risk of inducing acidosis in cases of placental insufficiency. Conversely, the Society for Maternal-Fetal Medicine (SMFM) in the United States recommends metformin as the first-line alternative to insulin in women with GDM when diet alone is insufficient in controlling hyperglycemia. Insulin is the most effective agent for hyperglycemia control due to its ability to be adjusted infinitely [20]. The United Kingdom’s National Institute for Health and Clinical Excellence (NICE) and the International College of Gynecology and Obstetrics (FIGO) consider oral antidiabetic drugs a viable option for women with low fasting glucose levels, as these medications are more likely to prevent hyperglycemia in such patients compared to those with high fasting glucose levels [2,21]. Recently, the Italian Drug Agency (Agenzia Italiana del Farmaco, AIFA) has updated the therapeutic indication for metformin, stating that its use during pregnancy and the periconceptional period may be considered if clinically appropriate, either in addition to or as an alternative to insulin therapy [22,23,24,25]. Moreover, the Italian Association of Medical Diabetologists (AMD), the Italian Society of Diabetology (SID) and the Italian Study Group of Diabetes in pregnancy have jointly published a position paper suggesting metformin as a therapeutic option for women with obesity, polycystic ovarian syndrome (PCOS), GDM, T2DM and those undergoing assisted reproductive technology (ART). However, the authors emphasize the necessity for further research, particularly regarding the long-term effects of fetal exposure to this drug [26].

Italian population-based studies on antidiabetic drug use in pregnancy are dated and limited to single regional experiences [27,28,29]. In this perspective, the AIFA has recently promoted the creation of a network, called MoM-Net (Monitoring Medication Use During Pregnancy—Network), which focuses on monitoring the use of different classes of medications in pregnancy, through the integration of different regional health databases.

The objective of this paper is to describe the prescription pattern of antidiabetic drugs in Italy and to evaluate its concordance with the Italian guideline on treatment of diabetes [4] to delineate potential targeted interventions to improve clinical practice.

## 2. Materials and Methods

This is a population-based study that relies on the record linkage of various regional health information flows that has allowed the identification of which antidiabetic drugs are prescribed to pregnant women in Italy. The main objective was to analyze the trend of drug prescriptions before, during and after pregnancy in women residing in eight Italian regions chosen for geographical representation (Lombardy, Veneto and Emilia-Romagna for the North; Tuscany, Umbria and Lazio for the Center; and Apulia and Sardinia for the South) and to evaluate the inter-regional variability in prescribing patterns to identify any critical issues related to prescribing appropriateness in relation to the Italian guideline on treatment of diabetes [4], with the ultimate aim of improving clinical practice.

The study population for this cross-sectional study was identified through a Common Data Model (CDM) based on three different data sources:-The Regional Birth Registry (Certificato di Assistenza al Parto, CeDAP), concerning sociodemographic characteristics of the mother and data regarding the pregnancy and the newborn;-The Demographic Database, from which administrative records were retrieved for those registered in the regional healthcare system;-The Drug Prescription Database, which includes all prescriptions reimbursed by the Italian National Healthcare Service, including date of dispensing, brand, active substance and number of packages prescribed.

A unique identification code was then created for each entry to link these data sources at a regional level anonymously. The Lazio Region designed the CDM and conducted the data analyses needed for its creation (Figure 1) [30,31]. Materials and methods were extensively reported in Belleudi et al., 2021 [32].

We selected 449,012 women, age ranging from 15 to 49 years old, from the eight Italian Regions who delivered in hospital between 1 April 2016 and 31 March 2018 (Table 1) [33]. We could not include voluntary and spontaneous abortions, as those data are not recorded in the CeDAP database.

For each patient, the date of pregnancy onset was estimated using the gestational age at birth. The study considered three periods: three trimesters before, three trimesters during and three trimesters after gestation.

The prevalence of medication use was calculated as the percentage of women with at least one drug prescription before, during and after pregnancy. Purposely, this study focused on the prevalence of antidiabetics use in each period. We defined the prevalence of drug use as “prevalent” when the medication was prescribed before conception and “incident” for new prescriptions occurring during pregnancy and the post-partum period.

The measure of exposure considered was the prevalence of use, defined as the percentage of women who have received at least one prescription within the specified period (the quantity of the drug was not taken into account). The data considered were related to the dispensing of the drug (purchase date at the pharmacy).

We also analyzed the shift to different antidiabetic categories (subgroups) and represented through a Sankey diagram the pattern of use during the different trimesters, in which the arrow’s width is proportional to the flow rate. This allowed us to display the proportion of women who changed their drug regimen during pregnancy. Similarly, the differences between antidiabetic prescriptions among different regions were assessed. All statistical analyses were performed using SAS (SAS Institute, Cary, NC, USA) and R (R Core Team, Vienna, Austria) version 9.4.

## 3. Results

The data describe a cohort of 449.012 women, age ranging from 15 to 49 years old, who had a pregnancy in the period between 1 April 2016 and 31 March 2018 in the eight participating regions, corresponding to 59% of Italian pregnancies.

The prevalence of antidiabetics prescription in the pre-conceptional period, representing cases of pre-existing DM, was 0.69%. This value rose to 2.64% during pregnancy, with a growing trend that reached 2.13% in the third trimester, in favor of insulins and its analogues (2%), and reduced to 0.49%1 in the postpartum period. The use of metformin halved over the course of pregnancy, going from 0.27% in the first trimester (47.87% of all antidiabetics) to 0.14% in the third trimester (6.52% of all antidiabetics) (Table 2).

Among users before pregnancy, oral antidiabetics were the most frequently prescribed drugs, with prevalence of use of 36%, 38.8% and 42% in the third, second and first preconceptional trimesters, respectively, due to the presence of type 2 DM. This was followed by insulins and analogues (25.9%, 25.3% and 25.3%), linked to type 2 and type 1 DM (Table 3).

While the users of insulin and analogues are mostly persistent in treatment in the trimesters preceding pregnancy, the users of other antidiabetic drugs are less constant. There is a marked increase in the prevalence of insulin use during pregnancy, ranging from 30.5% to 40.5% in the third trimester of gestation, due to an increased risk of GDM but also to the fact that insulin is of choice in pregnancy. After delivery, the prevalence of insulin use returned rapidly to pre-pregnancy levels. In turn, prevalence of oral antidiabetics use, which dropped to 8.5% in pregnancy, did not return to pregestational levels.

The high number of new users during pregnancy was likely due to the onset of GDM. In fact, the use in these women mostly occurred in the second and third trimesters and almost entirely in the category of insulins and analogues, with a prevalence of use of 0.92% and 2.02% in the second and third trimester, respectively. A small but not negligible share of new female users (up to 3.8% in the third trimester) used oral antidiabetics. Over 97% of these women no longer used these drugs after pregnancy.

We observed substantial regional variability in antidiabetic drugs prescriptions in the pre-conceptional period and in the first trimester of pregnancy. The largest increase in the prevalence of use of antidiabetic drugs occurred in the third trimester of pregnancy in all the regions considered. This increase was particularly marked in Tuscany (4.2%), Umbria (3.5%) and Emilia-Romagna (2.6%); instead, the lowest increases were observed in Lombardy (1.5%), Veneto (1.7%) and Apulia (1.7%) (Table 4).

The prevalence of use of antidiabetic drugs in the third trimester was significantly different among the regions considered (*p*-value < 0.001). The rate of antidiabetic prescription in Lombardy was 65% lower than in Tuscany (OR 0.35, 95% CI 0.33–0.37), 58% lower than in Umbria (OR 0.42, 95%CI 0.37–0.47) and 43% lower than in Emilia-Romagna (OR 0.57, 95% CI 0.53–0.61). Among the considered regions, the fraction of women older than 35 years old appeared to be comparable, being 37.2% in Lombardy, 37% in Veneto, 36.7% in Emilia-Romagna, 37.6% in Tuscany, 36.1% in Umbria, 41.7% in Lazio, 33.2% in Apulia and 42.8% in Sardinia.

We found that the use of antidiabetic drugs in the preconception period was almost double in the group of women with multiple pregnancies compared to those with single pregnancies (1.3% vs. 0.7%). The most commonly used medication in this group was metformin (Table 5). During pregnancy, there was an increase in prevalence in both groups, reaching 2.6% in women with single pregnancies and 3.4% in those with multiple pregnancies. This rise was almost entirely explained by a higher use of insulin. In the third trimester, insulin was used by 2.0% of women. The number of new users during the second and third trimesters increased from 1.1% to 2.0% for women with single pregnancies and from 0.7% to 1.8% for those with multiple pregnancies, then significantly reduced for both groups in the period following childbirth.

Finally, we compared the consumption of antidiabetics between the Italian population and the subgroup of foreign women living in Italy, in particular women coming from developed countries and women coming from countries with a high migratory pressure level [35]. In the three populations analyzed, no differences were observed in the consumption of antidiabetics in the pre-pregnancy period; on the contrary, in the second and third trimester of pregnancy, foreign women showed a doubled proportion of use compared to Italians (second trimester: 1.8% vs. 0.9%; third trimester: 3.6% vs. 1.8%). After delivery, consumption returned for all populations to the pre-pregnancy levels (Italians, 0.5%; women coming from developed countries, 0.1%; women coming from countries with high migratory pressure levels, 0.6%) (Figure 2).

## 4. Discussion

The prescribing profile of the various classes of antidiabetics in the Italian MoM-Net cohort seemed to be in line with the Italian guideline for the treatment of GDM which does not recommend the use of oral antidiabetics and non-insulin injection therapy during pregnancy [4].

We observed that the use of antidiabetics increased with pregnancy reaching its peak in the third trimester (2.13%) in favor of insulins and analogues, a trend which is assignable to the onset of GDM. These data can be compared with a recently published study describing the use of antidiabetic drugs before, during and after pregnancy in seven European regions over a period of time ranging from 2004 to 2010. The use appeared to be growing during pregnancy: the prevalence of use in the third trimester in the region with the highest prevalence was 2.2%, in line with the data observed in our sample; in the study, however, this value was more than double than the ones registered in the other studied regions [36]. This growing trend may be due both to the higher prevalence of diabetes in the population and to the older age of pregnant women.

Our analyses have shown that a small but not negligible share of new female users used oral antidiabetics (considering women in the third trimester, 2.13% assumed antidiabetics drugs of which up to 3.8% oral ones). This may indicate that, even if the overall prescription pattern seemed to be in line with the Italian guidelines for the treatment of GDM, metformin could have been used not only for T2DM but also for off-labels prescriptions and, consequently, that clinical practice is still heterogeneous [37,38]. The safety of metformin during pregnancy is still an open question, especially due to the lack of data on long-term outcomes in the offspring. Two retrospective population-based studies from New Zealand and Finland explored the long-term safety profile of metformin in the offspring, and no significant differences emerged in the outcomes evaluated [24,39]. Recently, a follow-up study was published examining children born to women with T2DM, both with and without exposure to metformin in utero, up to the age of 24 months. The study findings provided reassurance regarding the use of metformin during pregnancy for women with T2DM and its impact on the long-term health of their children [40]. Prior to this, a prospective, multicenter, international, randomized, parallel, double-masked, placebo-controlled trial found several maternal glycemic and neonatal adiposity benefits in the metformin group. This group experienced decreased maternal weight gain and insulin requirements, leading to enhanced glycemic control, a decrease in the number of large infants but an increase in the proportion of infants classified as small for gestational age [41]. Nevertheless, a recent review reported that, even if the mechanisms remain to be established, metformin is associated to catch-up growth and obesity during childhood, increasing the risk of future cardiometabolic diseases, but they once again highlighted the need for further investigations [42].

In line with this heterogeneity in practice observed, the assessment of the use of antidiabetics in the eight participating regions revealed a significant variability in the prescription pattern, especially in the third trimester of pregnancy. This variability may be due to different screening policies (OGTT 75 g screening prescribed on maternal risk factors or to all women between 24 and 28 gestational weeks [3,4]) and, consequently, to different therapeutic choices for glycemic control, as also shown by a recent US study [43].

We observed a significant regional variability in antidiabetic drugs prescriptions during pregnancy among the eight regions considered. Since the fraction of women older than 35 years old appeared to be comparable, this variability has to be likely referred to the already known different screening policies and therapeutic choices among the Italian regions.

From our analyses, it turned out that women with multiple pregnancies had a higher proportion of prescriptions of antidiabetic drugs compared to women with single pregnancies, both in the preconceptional period and during pregnancy. This ratio is likely linked to the older age of these women and consequently to a higher prevalence of T2DM. Additionally, we observed that women with multiple pregnancies more frequently used metformin in all the periods under consideration. This is probably due to the fact that metformin is recommended as the first choice for diabetes control in major guidelines and also aids in weight management. During pregnancy, there was an increase in prevalence of antidiabetic use in both groups, almost entirely explained by a higher use of insulin. In the third trimester, insulin was used by 2.0% of women. In fact, during the late stages of pregnancy, insulin is absorbed more slowly and may be less effective in lowering blood sugar levels, requiring larger doses. The use of insulin as opposed to oral therapy may be associated with an increased risk of hypertensive disorders of pregnancy and a higher likelihood of labor induction [44,45]. In confirmation of this, the number of new users during the second and third trimesters increased from 1.1% to 2.0% for women with single pregnancies and from 0.7% to 1.8% for those with multiple pregnancies, then significantly reduced for both groups in the period following childbirth.

By comparing the prevalence of use of the antidiabetics among Italian women and foreign women living in Italy coming from developed countries and from countries with a high migratory pressure level, we observed that the latter had a slightly higher proportion of antidiabetic prescriptions in all the studied periods, with a maximum increase during pregnancy. This trend is likely due to a genetic predisposition in developing GDM in women coming from countries with a high migratory pressure level, such as Afro-American, Hispanic, Asian and Native American women. This genetic predisposition to develop diabetes still stands when the woman has a low BMI [46]. A 2013 study highlighted differences in the prevalence of the use of antidiabetics according to the country of origin: for example, citizens of Sri Lanka and Bangladesh recorded more than four times higher rates than those of Kosovars, Moldovans and Romanians [47]. Moreover, women coming from countries with a high migratory pressure level struggled more than Italians in obtaining a clinical evaluation by a general practitioner or a specialized doctor, thus leading to a delay in adequate treatment initiation.

This study has some limitations, the main one being that we could not correlate the characteristics of the population at baseline with the type of diabetes occurred during pregnancy and its severity or the efficacy of the treatment. Also, we could not gather data regarding drug use in pregnancies which ended in a spontaneous or induced abortion, as these are not collected in the administrative databases to which we had access. Furthermore, no information on therapeutic indications and pregnancy outcomes for drug prescribing were available; consequently, we were not able to investigate the medication use patterns in more depth.

## 5. Conclusions

To date, the Italian MoM-Net cohort is the biggest and most representative Italian population-based study regarding the national prevalence of antidiabetic drugs use during pregnancy. Concerning the data presented in this work, the prescription pattern of antidiabetic drugs in Italy mostly includes medicines that are safe to take in pregnancy.

Given the limited information, the performed analyses provide an updated and exhaustive overview on antidiabetic drugs prescription pattern in Italian pregnant women, which can help identify critical aspects in the management of diabetes during pregnancy in Italy.

We believe that further descriptive studies led by the MoM-Net group and coordinated at a national level could be successful in improving the current Italian clinical practice regarding treatment choices in pregnancy while also playing an important role in promoting standardization of the prescriptions between the different regions.

## Figures and Tables

**Figure 1 ijerph-20-07139-f001:**
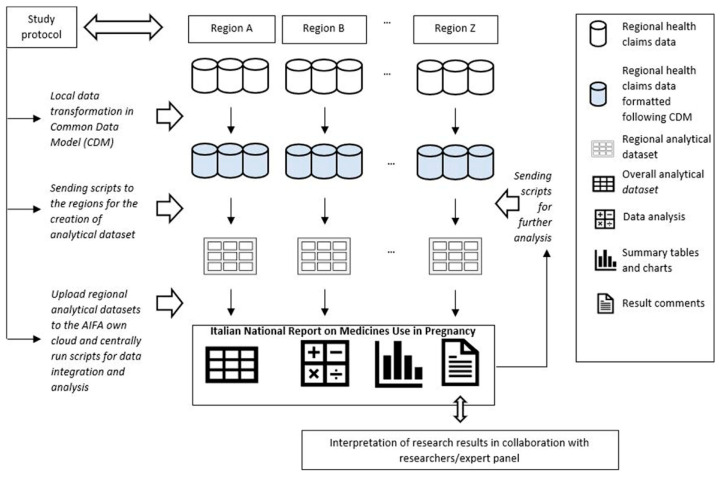
Analytical approach used to execute observational studies within MoM-Net. Reprinted from Ref. [32].

**Figure 2 ijerph-20-07139-f002:**
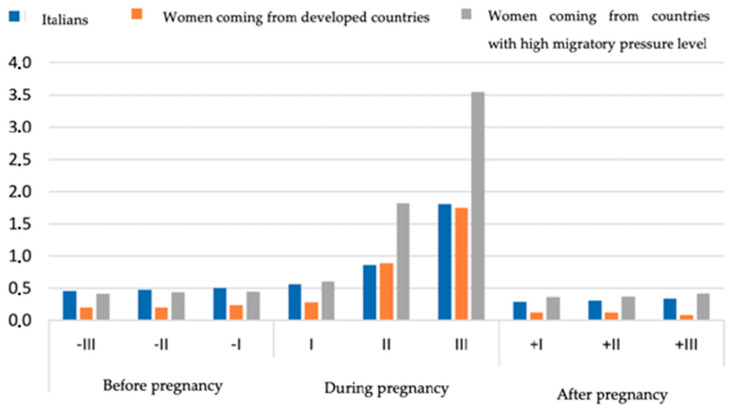
Prevalence of antidiabetic use before, during and after pregnancy.

**Table 1 ijerph-20-07139-t001:** Study cohort characteristics (n = 449,012). Reprinted from Ref. [33].

	n	%
**Age group**		
≤24	33,651	7.5
25–29	92,333	20.6
30–34	154,588	34.4
35–39	124,680	27.8
≥40	43,760	9.7
Of which ≥45	3438	7.9
**Nationality**		
Italian	358,467	79.8
Foreign	88,629	19.8
Low-income countries *	86,159	97.7
High-income countries *	2470	2.3
**Level of education**		
None/primary school	106,759	23.8
Secondary school	200,618	44.7
Bachelor’s degree/post-bachelor’s degree	139,559	31.1
Missing	2076	0.5
**Occupational status**		
Employed	284,069	63.3
Unemployed/Looking for first job	54,492	12.1
Housewife	98,450	21.9
Other	7210	1.6
Missing	4791	1.1
**Previous delivery**		
No	227,525	50.7
Yes	221,487	49.3
Of which cesarean section	59,782	27.0
**Previous spontaneous abortions ****		
0	360,619	80.3
1	65,997	14.7
2	22,396	5.0
**Gestational age**		
Preterm delivery (<37 weeks)	30,774	6.9
Term delivery (37–41 weeks)	415,366	92.5
Post-term delivery (>41 weeks)	2872	0.6
**Parity**		
1	440,765	98.2
2+	8247	1.8
**Invasive prenatal diagnosis**		
None	394,785	88.1
Chorionic villus sampling	20,435	4.6
Amniocentesis	31,423	7.0
Other invasive test	1433	0.3
**Medically assisted procreation *****		
No/Not classified	360,558	97.0
Yes	11,233	3.0
**Cesarean section**		
No	312,785	69.7
Yes	136,227	30.3

* The following countries were considered Advanced Development Countries: Andorra, Australia, Austria, Belgium, Canada, the Vatican City, South Korea (under discussion), Denmark, Finland, France, Germany, Japan, Greece, Ireland, Iceland, Israel, Italy, Liechtenstein, Luxembourg, Monaco, Norway, New Zealand, the Netherlands, Portugal, the United Kingdom, San Marino, Spain, the United States, Sweden and Switzerland. Countries with High Migration Pressure were considered those in Central and Eastern Europe (including those belonging to the European Union) and Malta, countries in Africa, Asia (excluding South Korea, Israel and Japan), Central and South America and Oceania (excluding Australia and New Zealand) [34]. ** Lazio took into account the number of voluntary and spontaneous abortions. *** Data from Lazio and Umbria are not included.

**Table 2 ijerph-20-07139-t002:** Women with at least one antidiabetic prescription in the trimesters before, during and after pregnancy.

	Trimester BEFORE Pregnancy						Trimester DURING Pregnancy						Trimester AFTER Pregnancy					
	−III		−II		−I		I		II		III		+I		+II		+III	
	n	%	n	%	n	%	n	%	n	%	n	%	n	%	n	%	n	%
**Antidiabetics**	2012	0.45	2094	0.47	2204	0.49	2540	0.57	4682	1.04	9541	2.13	1369	0.3	1451	0.32	1595	0.36
Oral hypoglycemics	58	0.01	66	0.01	63	0.01	46	0.01	24	0.01	11	0	28	0.01	30	0.01	43	0.01
GLP-1 analogues	13	0	15	0	19	0	10	0	2	0	4	0	4	0	10	0	22	0
Glyphozines (alone or combined)	11	0	10	0	19	0	9	0	0	0	1	0	3	0	12	0	22	0
Gliptins (DPP-4 inhibitors alone or combined)	24	0.01	26	0.01	23	0.01	24	0.01	3	0	3	0	12	0	16	0	19	0
Insulins and analogues	877	0.2	857	0.19	848	0.19	1497	0.33	4151	0.92	9027	2.02	1006	0.22	943	0.21	970	0.22
Metformin	1143	0.25	1242	0.28	1351	0.3	1216	0.27	581	0.13	622	0.14	375	0.08	525	0.12	637	0.14
Pioglitazone (alone or combined)	12	0	11	0	11	0	9	0	5	0	1	0	5	0	7	0	8	0
Repaglinide	9	0	10	0	10	0	9	0	3	0	1	0	2	0	2	0	3	0

**Table 3 ijerph-20-07139-t003:** Pattern of use of the different classes of antidiabetics in the trimesters before, during and after pregnancy.

	Trimester BEFORE Pregnancy						Trimester DURING Pregnancy						Trimester AFTER Pregnancy					
	−III		−II		−I		I		II		III		+I		+II		+III	
	n	%	n	%	n	%	n	%	n	%	n	%	n	%	n	%	n	%
**Main users N = 3103**																		
Other (antidiabetic)	55	1.8	63	2	65	2.1	35	1.1	6	0.2	5	0.2	9	0.3	11	0.4	23	0.7
New antidiabetics	34	1.1	38	1.2	48	1.5	29	0.9	2	0.1	2	0.1	9	0.3	22	0.7	39	1.3
Oral antidiabetics	1117	36	1205	38.8	1303	42	902	29.1	349	11.2	263	8.5	239	7.7	338	10.9	376	12.1
Insulins and analogues	805	25.9	786	25.3	785	25.3	946	30.5	1294	41.7	1257	40.5	807	26	815	26.3	804	25.9
Not users	1092	35.2	1011	32.6	902	29.1	1191	38.4	1452	46.8	1576	50.8	2039	65.7	1917	61.8	1861	60
**New users in pregnancy N = 9621**																		
Other (antidiabetic)	0	0	0	0	0	0	14	0.1	15	0.2	4	0	3	0	5	0.1	7	0.1
New antidiabetics	0	0	0	0	0	0	7	0.1	3	0	5	0.1	0	0	4	0	9	0.1
Oral antidiabetics	0	0	0	0	0	0	283	2.9	235	2.4	361	3.8	75	0.8	96	1	117	1.2
Insulins and analogues	0	0	0	0	0	0	323	3.4	2778	28.9	7643	79.4	118	1.2	57	0.6	78	0.8
Not users	9621	100	9621	100	9621	100	8994	93.5	6590	68.5	1608	16.7	9425	98	9459	98.3	9410	97.8

**Table 4 ijerph-20-07139-t004:** Women with at least one antidiabetic prescription in the trimesters before, during and after pregnancy by region.

Regions	Trimester BEFORE Pregnancy						Trimester DURING Pregnancy						Trimester AFTER Pregnancy					
	−III		−II		−I		I		II		III		+I		+II		+III	
	n	%	n	%	n	%	n	%	n	%	n	%	n	%	n	%	n	%
**Antidiabetics**																		
Lombardy	349	0.3	359	0.3	356	0.3	442	0.3	1.065	0.8	2.132	1.5	301	0.2	311	0.2	351	0.3
Veneto	152	0.2	159	0.3	166	0.3	210	0.3	549	0.9	1.034	1.7	149	0.2	153	0.2	171	0.3
Emilia Romagna	226	0.4	242	0.4	251	0.4	310	0.6	749	1.3	1.452	2.6	180	0.3	197	0.3	202	0.4
Tuscany	246	0.5	235	0.5	262	0.6	302	0.6	746	1.6	1.980	4.2	136	0.3	142	0.3	152	0.3
Umbria	34	0.3	33	0.3	40	0.4	44	0.5	132	1.4	338	3.5	27	0.3	25	0.3	27	0.3
Lazio	489	0.7	530	0.8	572	0.8	608	0.9	694	1.0	1.386	2.1	260	0.4	289	0.4	318	0.5
Apulia	382	0.7	393	0.8	419	0.8	473	0.9	515	1.0	858	1.7	206	0.4	219	0.4	253	0.5
Sardinia	134	0.9	143	0.9	138	0.9	151	1.0	232	1.5	361	2.3	110	0.7	115	0.7	121	0.8

**Table 5 ijerph-20-07139-t005:** Women with single pregnancies and multiple pregnancies with at least one antidiabetic prescription in the trimesters before, during and after pregnancy.

	Trimester BEFORE Pregnancy						Trimester DURING Pregnancy						Trimester AFTER Pregnancy					
	−III		−II		−I		I		II		III		+I		+II		+III	
	n	%	n	%	n	%	n	%	n	%	n	%	n	%	n	%	n	%
**Single pregnancies**																		
**Antidiabetics**	1953	0.44	2025	0.46	2127	0.48	2472	0.56	4549	1.03	9351	2.13	1343	0.30	1420	0.32	1562	0.35
Oral hypoglycemics	57	0.01	63	0.01	63	0.01	46	0.01	23	0.01	11	0.00	27	0.01	29	0.01	41	0.01
GLP-1 analogues	13	0.00	14	0.00	19	0.00	10	0.00	2	0.00	4	0.00	4	0.00	10	0.00	22	0.01
Glyphozines (alone or combined)	9	0.00	8	0.00	16	0.00	8	0.00	0	0.00	1	0.00	3	0.00	12	0.00	22	0.01
Gliptins (DDP-4 inhibitors alone or combined)	24	0.01	26	0.01	23	0.01	24	0.01	3	0.00	3	0.00	12	0.00	16	0.00	19	0.00
Insulins and analogues	863	0.20	843	0.19	835	0.19	1.472	0.33	4.039	0.92	8.855	2.01	992	0.23	927	0.21	955	0.22
Metformin	1097	0.25	1188	0.27	1286	0.29	1169	0.27	555	0.13	597	0.14	362	0.08	509	0.12	618	0.14
Pioglitazone (alone or combined)	12	0.00	11	0.00	11	0.00	9	0.00	5	0.00	1	0.00	5	0.00	7	0.00	8	0.00
Repaglinide	8	0.00	9	0.00	10	0.00	9	0.00	3	0.00	1	0.00	2	0.00	2	0.00	3	0.00
**Multiple pregnancies**																		
**Antidiabetics**	59	0.72	69	0.84	77	0.93	68	0.82	133	1.61	190	2.36	26	0.32	31	0.38	33	0.40
Oral hypoglycemics	1	0.02	3	0.05	0	0.00	0	0.00	1	0.02	0	0.00	1	0.02	1	0.02	2	0.03
GLP-1 analogues	0	0.00	1	0.11	0	0.00	0	0.00	0	0.00	0	0.00	0	0.00	0	0.00	0	0.00
Glyphozines (alone or combined)	2	0.09	2	0.09	3	0.13	1	0.04	0	0.00	0	0.00	0	0.00	0	0.00	0	0.00
Gliptins (DDP-4 inhibitors alone or combined)	0	0.00	0	0.00	0	0.00	0	0.00	0	0.00	0	0.00	0	0.00	0	0.00	0	0.00
Insulins and analogues	14	0.17	14	0.17	13	0.16	25	0.30	112	1.36	172	2.14	14	0.17	16	0.19	15	0.18
Metformin	46	0.57	54	0.67	65	0.80	47	0.58	26	0.32	25	0.32	13	0.16	16	0.20	19	0.24
Pioglitazone (alone or combined)	0	0.00	0	0.00	0	0.00	0	0.00	0	0.00	0	0.00	0	0.00	0	0.00	0	0.00
Repaglinide	1	0.05	1	0.05	0	0.00	0	0.00	0	0.00	0	0.00	0	0.00	0	0.00	0	0.00

## Data Availability

The data that support the findings of this study are available from the Italian regions participating to MoM-Net group (Lombardy, Veneto, Emilia Romagna, Tuscany, Umbria, Lazio, Puglia, Sardinia), but restrictions apply to the availability of these data, which were used under license (as by third-party sources) for the current study, and so are not publicly available. However, data are available from the authors with permission of the Italian regions, which are the data owners.

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
