# Peer review of "Antidiabetic Therapy during Pregnancy: The Prescription Pattern in Italy"

_ijerph, 2023, doi:10.3390/ijerph20237139_

Round 1
Reviewer 1 Report
Comments and Suggestions for Authors
Thank you very much for the opportunity to review this article about Antidiabetic Therapy during Pregnancy: The Prescription Pattern in Italy. Overall, I found it to be a bold article that offers a different approach and, therefore, may provide a valuable perspective for clinical practice.
However, some minor aspects are amenable to revision. These are as follows:
I.- After a calm reading, I’m not able to identify the objectives of the manuscript: "to highlight the prescription pattern of antidiabetics before, during and after pregnancy in Italy and to evaluate its appropriateness".
II.- In the Methods section, I propose that at the beginning, they describe in a general way the type of study being performed. Then, the eligible population and its characteristics. The latter, together with the selection criteria, is essential to be well described since one of the main problems of this type of study lies precisely in the potential for selection bias.
Author Response
Thank you for the helpful suggestions as to how we might further improve the paper. Below we respond to each suggestion with details about how we amended the manuscript

Reviewer 2 Report
Comments and Suggestions for Authors
Thank you for the opportunity to review this drug utilisation study in Italy, with focus on drugs for diabetes treatment in pregnancy.
The Introduction is mainly focussed on different clinical treatment guidelines of gestational diabetes mellitus (GDM), in Europe and outside Europe. However, the study - as fas as I can see - extract drug use data for any type of diabetes, not only GDM. I encourage the authors to re-structure the Introduction in a more balanced manner, that is indicate the prevalence of T1D and T2D in women of reproductive age (as these women will have diabetes once they enter the pregnancy period), on top of estimates for GDM.
The Introduction could also be improved by focussing more on the extent of use of anti diabetic medication in pregnancy in prior published research, possibly by type of diabetes, and whether prior studies have examined same question in Italy. In brief, I would like to understand what this study adds to prior research and knowledge, and what the authors did to overcome limitations of prior studies. To my knowledge, there are published drug utilisation studies in Italy, and their results in relation to diabetes drugs (if any) should be mentioned in the Introduction.
Aims: I find the aims not fully reflecting what is presented in the Results, eg the region variations, the analysis by nationality. Could the authors have a more specific aim that reflects what they actually aimed at examining? The part of the aim assessing the appropriateness of prescribing is not examined as far as I can see. In the methods, I could not find a description on how this examination of appropriate prescribing was conducted.
Methods: For reproducibility purposes, please expand the methods section, specifically for: (1) how the trimester windows were defined; this is important, as the length of the various trimesters as number of days varies according to different definitions; (2) how the authors defined exposure to an antidiabetic medication in each window. For example, did they require that the date of the filled prescription fell within the length of an exposure window, or did they also consider the number of tables/unit dispensed in the latest prescription and whether the day supply overlapped with the windows of exposure? Point 2 is critical in this setting, as the study is dealing with long-term medication exposures, and it is expected that one prescription fill covers up to 3 months of treatment.
The distinction between prevalent and incident users should be expanded with more details, and clearly indicate how these were defined (also in view of my prior comment). For instance, was incident use only defined in pregnancy, or also postpartum? GDM has also onset in mid-pregnancy, and elevated blood glucose in first trimester is often the result of an unknown pre-gestational diabetes in the woman. Could the authors clarify/expand how they addressed this point, since the aim of the study was also to evaluate the appropriateness of prescriptions? The start of the Discussion mentions this point, but again the method and process used for this assessment is not described in the methods, and also not clearly indicated in the Results.
Demographics of the sample: can the authors better describe how information on sociodemographic are collected? Looking at Table 1, I see many covariates listed, but the methods does not provide any information on how these are collected and measured. For instance, "nationality": I would like to know (again for reproducibility purposes) how high and low-income countries for foreign mothers were classified. I find surprising that 97.7% of foreign mothers resident in Italy are from high income countries, given the high immigration rate into Italy from less developed countries.
I would have appreciated some information about screening for GDM in Italy, overall or by region if specific differences in screening practice are known to the authors. Is GDM screening universal on Italy, or targeted? That is a main component driving the prescribing for GDM treatment, that remains unclear.
Was it possible to extract the refund/reimbursement code to identify the underlying indication for use of the anti-diabetes drugs? It would be a strength for the study to know which condition these women had before entering the pregnancy and during. The authors acknowledge this limitation, but I was wondering why access to the refund codes is not possible in the data extracts per region.
It would have been interesting to know how many women were prescribed more than 1 antidiabetic drug (switching patterns for instance across drug groups). By looking at the table 3, it seems that the groups of drugs are mutually-exclusive, am I correct? However, some women may discuss with their doctor switch to a more favourable medication in pregnancy, especially those with T2D already before pregnancy. This aspect of how drug use across the various groups was operationalised should also be better explained in the Methods, because it remains unclear.
One important limitation of the study is the coverage of regions in Italy that are mainly representing the North and Central Italy. Only Puglia is included from the South of Italy. I think the authors need to acknowledge this point and what results one would expect with inclusion of southern regions of Italy in this study. This is important given the different distribution of obesity, access to medical care, education, etc between regions/areas in Italy.
The conclusion and abstract should again reflect the actual analysis of drugs for diabetes; I am not clear why the authors focus only on GDM. If that was the focus, I feel that the results should have been presented differently and only for new incident users of antidiabetic drugs in 2nd/3rd trimester.
Author Response

(The authors gave the same response as above.)

Round 2
Reviewer 2 Report
Comments and Suggestions for Authors
Dear authors, thank you for revising the manuscript and considering the points/questions raised in the first round. It is important that some aspects are detailed in the manuscript, not only in the response letter. Please amend the Methods sections as requested in the first round of review concerning the following points:
IV-2: how you define exposure, that is that you do not consider dispensed supply of the drug in the end of one trimester overlapping the following trimester in the exposure definition; this is unusual method in perinatal pharmcoepidmeiology examining these types of medications; you need to describe this in the methods for reproducible research.
V: same as above; I cannot see a clear description of the prevalent and incident users in the methods, please be clear.
VI: the categorisation of countries must again be explained in the methods, for reproducibility of research.
Thank you for amending the methods, so that other researchers can be informed in details on how you defined and processed the data.
Comments on the Quality of English Language
-
Author Response
We thank the Referee for the suggestions on how to improve reproducibility of our research work. The requested details of the methods have now been specified in the Methods section, they are highlighted in light blue: - IV-2: the definition of exposure can now be found in lines 183-186 of the Material and Methods section; - V: the definition of prevalence and incident was already reported in lines 178-182 of the Material and Methods section. We have now added that "incident" was used also for new prescriptions in the post-partum period; - VI: we have now specified the categorisation of countries in line 166-173 of the Material and Methods section.
